# A Lightweight, Centralized, Collaborative, Truncated Signed Distance Function-Based Dense Simultaneous Localization and Mapping System for Multiple Mobile Vehicles

**DOI:** 10.3390/s24227297

**Published:** 2024-11-15

**Authors:** Haohua Que, Haojia Gao, Weihao Shan, Xinghua Yang, Rong Zhao

**Affiliations:** 1College of Science, Beijing Forestry University, Beijing 100083, China; qh13005968844@bjfu.edu.cn; 2Department of Fan Gongxiu Honors College, Beijing University of Technology, Beijing 100124, China; gaohaojia@emails.bjut.edu.cn; 3Department of Electronic Engineering, Tsinghua University, Beijing 100084, China; shanwh22@mails.tsinghua.edu.cn; 4School of Computer Science and Technology, North University of China, Taiyuan 030051, China

**Keywords:** SLAM, lightweight system, centralized collaborative, TSDF, mobile robot, visual inertial odometry

## Abstract

Simultaneous Localization And Mapping (SLAM) algorithms play a critical role in autonomous exploration tasks requiring mobile robots to autonomously explore and gather information in unknown or hazardous environments where human access may be difficult or dangerous. However, due to the resource-constrained nature of mobile robots, they are hindered from performing long-term and large-scale tasks. In this paper, we propose an efficient multi-robot dense SLAM system that utilizes a centralized structure to alleviate the computational and memory burdens on the agents (i.e. mobile robots). To enable real-time dense mapping of the agent, we design a lightweight and accurate dense mapping method. On the server, to find correct loop closure inliers, we design a novel loop closure detection method based on both visual and dense geometric information. To correct the drifted poses of the agents, we integrate the dense geometric information along with the trajectory information into a multi-robot pose graph optimization problem. Experiments based on pre-recorded datasets have demonstrated our system’s efficiency and accuracy. Real-world online deployment of our system on the mobile vehicles achieved a dense mapping update rate of ∼14 frames per second (fps), a onboard mapping RAM usage of ∼3.4%, and a bandwidth usage of ∼302 KB/s with a Jetson Xavier NX.

## 1. Introduction

Autonomous exploration enables robots to autonomously explore, discover, and gather information in environments where direct human intervention may be impractical, unsafe, or impossible. In an autonomous exploration task, Simultaneous Localization And Mapping (SLAM) algorithms provide the necessary spatial awareness for the robot to navigate and build a map of its surroundings in real-time, and they have been a focus of robotics research over the past few decades [1,2,3]. Among them, visual-based SLAM systems have become very popular due to their advantages of low weight, low power, low cost, and the ability to provide rich information about the environment. These advantages make visual-based systems highly suitable for resource-constrained platforms, such as Automated Ground Vehicles (AGVs) and Unmanned Aerial Vehicles (UAVs), which are ideal for autonomous exploration tasks due to their agility and small size.

Many successful visual-based SLAM systems represent the world by converting input images into a set of feature points. This efficient process yields robust and real-time localization and mapping results [4,5]. Although feature-based SLAM systems demonstrate centimeter-level localization accuracy, they map the world as a collection of 3D sparse points. This insufficient representation limits their use for high-level tasks such as obstacle avoidance and path planning. Truncated Signed Distance Function (TSDF) has recently proven to be an effective visual-based implicit representation for constructing a more geometrically complete environment [6], also demonstrating its great versatility in other robotic applications [7,8].

Although TSDF-based dense SLAM systems for single robots have reached a certain level of maturity and robustness [1,9], they often encounter an unavoidable problem: robots must operate on resource-constrained platforms. This limitation prevents robots from operating for an extended period of time over larger areas, as the computational time, memory footprint, and battery life are bounded by the resources of the robot. Aiming to tackle this problem, multi-robot collaborative SLAM has become a solution. Multi-robot collaborative SLAM systems deploy multiple robots in a large-scale environment, dividing the scenario into smaller areas and allowing different robots to map distinct regions. This not only alleviates the computation and memory pressure on a single robot but also enhances the efficiency and robustness of the mission through shared information. However, the majority of existing multi-robot SLAM systems represent the world using 3D sparse landmarks [10,11], with few addressing the challenge of collaborative dense SLAM. Another issue with existing works is their lack of practicality, as they often rely on pre-recorded datasets for real-world simulations or employ heavyweight platforms to compensate for computation time and memory storage. This hinders robots from effectively carrying out real-world missions, such as search-and-rescue and cave exploration tasks, where resource-constrained small-sized platforms are needed and factors like communication range and a limited bandwidth must be taken into consideration.

To address the aforementioned problems, we propose an efficient and robust multi-robot collaborative dense SLAM system. Inspired by CCM-SLAM [10], our system is built under a centralized architecture, efficiently outsourcing computationally expensive tasks from agents to a ground station (server) while ensuring that all tasks critical to the autonomy of each agent are still run onboard. We extend the decentralized dense multi-robot SLAM framework from Dubois et al. [12] to a centralized system by utilizing TSDF submaps. Rather than directly aligning TSDF submaps to find loop closures as in [12], we incorporate the visual-based place recognition method [13] along with TSDF submaps to find correct loop closure inliers. For submap matching, different from the Iterative Closest Point [14] (ICP)-based method proposed in [12], we adopt the lightweight correspondence-free submap matching method proposed in the work of Voxgraph [9] to maintain global consistency in real-time. In addition, to further ease the computation pressure on the agent, we have designed a lightweight and accurate TSDF-based dense mapping method based on the lightweight TSDF integration method proposed in the work of [15] and the non-projective TSDF integraion method proposed in the work of [16]. Experiments with both datasets and real-world scenarios demonstrate the efficiency, lightweightness, accuracy, and robustness of our proposed system.

The main contributions of this work are as follows:We present a centralized collaborative dense mapping system based on TSDF submaps, alleviating computation and memory pressure on mobile vehicles. Real-world experiments show the applicability and robustness of our system.We provide a lightweight and accurate TSDF mapping method to enable real-time and precise 3D reconstruction on resource-constrained mobile vehicles.We descriibe a robust and accurate loop closure detection method that rejects loop closure outliers through a combination of keyframe-based and TSDF-based methods.We integrate a lightweight submap matching method [9] into a centralized multi-robot pose graph optimization problem to enable real-time global consistency.

## 2. Related Work

### 2.1. Dense Single-Robot SLAM

Dense SLAM systems have employed various map representations to construct a more geometrically complete map compared to feature-based SLAM systems [4,5]. Newcombe et al. [17] proposed DTAM, a fully direct system that works with all the raw pixel information and estimate depth values based on photometric errors. Engel et al. [18] preformed a direct semidense reconstruction utilizing pixels with strong gradients (i.e., edges) along with keyframes. Whelan et al. [19] represented the world as a collection of surfels through non-rigid surface deformations. Most recently, a neural network-based 3D reconstruction method, neural radiance field (NeRF) [20], has attracted significant attention. NeRF-based SLAM systems [21,22] use pretrained or online-trained approaches to enable detailed mapping results. While these systems observe the world with denser representations, they lack the obstacle and free space information in the maps, which is crucial for autonomous exploration tasks. In contrast, volumetric maps represent the world as a collection of voxels, storing information about the occupied or free status within them.

TSDF-based implicit dense mapping is a volumetric mapping approach [23] that has demonstrated compelling results recently [6]. Such representation has the ability to incrementally fuse noisy sensor data from a consumer-grade depth camera and provides subvoxel resolutions to reconstruct a more accurate surface. Furthermore, to enable real-time operations on low-grade robotic platforms, Oleynikova et al. [1] proposed Voxblox, a systematic approach providing accurate real-time TSDF integration on CPU for relatively large voxels. Based on Voxblox, Voxfield [16] uses a novel non-projective TSDF formulation method to correct the projective signed distance error for each voxel from Voxblox. However, both Voxblox [1] and Voxfield [16] update all the free-space voxels along every ray in each frame, leading to redundant voxel updates, as different rays may intersect each other. This redundant calculation made it challenging to deploy TSDF-based algorithms on resource-constrained micro vehicles to achieve real-time performance.

### 2.2. Dense Multi-Robot SLAM

Existing dense multi-robot SLAM systems can be divided into two major categories: decentralized or centralized. For decentralized systems, Schuster et al. [24] proposed a multi-robot stereo-visual dense SLAM system based on pointcloud submaps. The system’s global consistency is maintained by ICP alignment of each submap and pose graph optimization. Similar to [24], Dubois et al. [12] used TSDF submaps to generate surface polygonal meshes and extract point clouds from the meshes to perform ICP matching to find loop closures. Based on the submap matching results, they also formulated a pose graph optimization problem to maintain global consistency. Kimera-Multi [25] proposed another decentralized system for metric semantic dense SLAM, extending Kimera’s method [26] to a multi-robot version. Although the aforementioned decentralized systems yielded great results, they inevitably added more computation and memory pressure to the robots (e.g. loop closure detection, pose graph optimization).

Centralized systems seek to alleviate the aforementioned limitations by transferring non-time-critical, memory-heavy, and computationally expensive processes to a central server. Bartolomei et al. [27] proposed a centralized dense mapping system utilizing external GPS information. The server collects keyframes and point clouds from each agent and perform global pose graph optimization and global map fusion. However, it relies on GPS information to coordinate each agent. CVIDS [28] is another centralized dense mapping system. The agents send monocular images to the server, and the server performs depth estimation and global TSDF fusion. However, it does not maintain a dense map on the agent side, which prevents the agent from performing high-level tasks such as obstacle avoidance. In this work, we extend the work of Dubois et al. [12] to a centralized system. The agent performs lightweight TSDF map reconstruction and send the TSDF submaps to the server. The server performs loop closure detection and lightweight pose graph optimization based on keyframes, trajectories, and TSDF submaps.

## 3. Methods

### 3.1. System Overview

The architecture of our proposed dense multi-robot SLAM system is shown in Figure 1. We use a centralized multi-robot framework to ease the computation and memory pressure of robotic agents by offloading non-time-critical, memory-heavy, and computationally expensive tasks to the server while ensuring the basic autonomy of each agent. For basic autonomy, each agent runs a real-time visual inertial odometry (VIO) module to estimate its pose and simultaneously runs a TSDF mapping module to generate a dense map for high-level tasks such as obstacle avoidance. Note that both the VIO module and TSDF mapping module on board the agent only keep the memory of its vicinity. This not only reduces the size of the bundle adjustment (BA) optimization in the VIO module but also decreases the size of voxel updates in the TSDF mapping module. This process massively reduce the memory and computational pressure on resource-constrained robotic agents. Although the agent only keeps the memory of its vicinity, it continuously sends necessary data (keyframes from the VIO module and TSDF submaps from the TSDF mapping module) to the server to offload information, where the server acts as a bookkeeper to store all of the information from each agent. Note that the agent’s autonomy is independent of the server, because even in the case of a complete loss of connection to the server, the agent can still run local VIO and the dense mapping module to ensure its autonomy.

In addition to bookkeeping in the server map stack, the server also runs place recognition, global Pose Graph Optimization (PGO), and global map fusion modules. When performing coordination, the server does not acknowledge any prior information of the initial locations of agents. Each agent’s map in the server maintains a local coordinate frame and is independent of each other. By continuously detecting overlapping areas (i.e., loop closure detection) between agents in the place recognition module, the server identifies correlations among agents and sends that constraint information to the global PGO module. The global PGO module will then correct the drifted pose of each agent based on the relative poses and TSDF submaps. After the global PGO, the local coordinate frames of the agents will be fused into one global coordinate frame, and the maps corresponding to each agent will be merged into one global map.

### 3.2. System Modules

The key modules of the proposed system shown in Figure 1 are described in detail below.

#### 3.2.1. Local TSDF Mapping

For dense mapping, the agent processes incoming raw pointcloud measurements with the associated poses provided by the VIO module [29] and incrementally builds a local voxel map of its vicinity. In order to compensate for the projective signed distance error from raycasting [1], we use the non-projective signed distance [16] to improve the mapping accuracy. The TSDF map is constructed using a set of spatially hashed voxels $V_{i}$ with predefined voxel size $\nu \in R^{+}$. Each voxel $V_{i}$ has a global index $v_{i}\in Z^{3}$, and the center position of each voxel $V_{i}$ is represented by $x_{i}=\nu v_{i}\in R^{3}$. In addition, each voxel $V_{i}$ stores the signed distance $D_{i}$, the weight value $W_{i}$, and the normalized gradient $g_{i}\in R^{3}$.

To update the TSDF map, at each local frame *k*, we first need to cast a ray from the sensor origin $s_{k}\in R^{3}$ to the measured surface point $p_{j}$ to compute the projective signed distance at every voxel along this ray: (1)$$d_{p}=\mathrm{sign}\left(\left(p_{j}-s_{k}\right)\cdot \left(p_{j}-x_{i}\right)\right)∥p_{j}-x_{i}∥$$

A key process to further compute the non-projective signed distance $d_{np}$ based on the projective signed distance $d_{p}$ is the computation of the normalized gradient of each voxel $g_{i}$.The gradient $g_{i}$ of each voxel is approximated by the surfacepoints normal vectors ${\left\{n\right\}}_{k}\in R^{3}$. Utilizing the gradient $g_{i}$ information, the non-projective signed distance $d_{np}$ is computed using the geometric relationship between the ray, surface normal, and the gradient. For more details, please refer to [16]. Finally, the voxels can be updated as follows: (2)$$D_{i}\leftarrow \frac{W_{i}D_{i}+w_{ijk}d_{np}}{W_{i}+w_{ijk}}$$
(3)$$W_{i}\leftarrow \min\left(W_{i}+w_{ijk},W_{\max}\right)$$
where we adopt the weight $w_{ijk}$ definition as in [16], and the non-projective signed distance $d_{np}$ is truncated at a distance of 3ν.

Such incremental refinement ensures the local consistency of the TSDF submap, and the grouped raycasting method, first proposed in [1] and adopted in [16], enables real-time updating on CPU for relatively large voxels. However, the inherent process of explicitly updating all the free-space voxels both in [1] and ref. [16] leads to a increased TSDF computation time. This prevents the resource-constrained platforms from performing real-time TSDF mapping. To update the free-space voxels more effectively, we terminate the raycasting process early based on subvoxel-based points. This process is summarized in Algorithm 1. There are two cases in which we terminate the raycasting process early. In the first case, to reduce the density of the points in the voxel, we divide the voxel into 8 subvoxels. For each subvoxel, we only insert one point. Once the point is inserted, we mark the subvoxel as occupied. The other point that has the same location with the occupied subvoxel will be discarded. This subsampling process reduces the number of points that need to be raycast, resulting in increased efficiency. In the second case, we performed a ray collision check. For the non-occupied subvoxels, we cast a ray from the point to the sensor origin. Before updating the voxels along this ray, we count how many rays have passed through each voxel. If the voxel has been passed through more than three times, we terminate the ray and discard the other voxels along this ray. Note that because we performed the raycasting from the point to the sensor origin, the rays will draw together near the sensor origin, and before executing the ray collision check, the vast majority of the free-space voxels will be updated at least once. By combining these two checking processes, the computation time of TSDF is greatly reduced, enabling real-time dense TSDF mapping on resource-constrained platforms.
**Algorithm 1** Lightweight TSDF Integration**Require:** Sensor origin sk, pointcloud of current scan *p*, voxel indexes *v***Ensure:** Updated voxel state
 1:**for** each point pj in *p* **do** 2:    SearchForVoxelIndex(pj,vj) 3:    **if** IsSubVoxelOccupied(vj) **then** 4:        **continue** 5:    **end if** 6:    SetSubVoxelAsOccupied(vj) 7:    CastRayFromPointToOrigin(sk,pj,v) 8:    **for** each voxel index vi along the ray in *v* **do** 9:        **if** VoxelGotRayCollision(vi) **then**10:           **break**11:        **end if**12:        SetVoxelRayCollisionStatus(vi)13:        UpdateTSDFVoxel(vi)14:    **end for**15:**end for**


#### 3.2.2. Loop Closure Detection

The inter-robot localization method is different from [12], which only used the SDF submaps to perform a time-consuming ICP-based loop detection. We first perform fast loop detection by utilizing the visual information from the keyframes in the VIO module to compute an initial transformation $T_{ij}$ between agent *i* and agent *j*. Then, based on the initial transformation, we utilize the dense geometric information of SDF submaps to reject loop closure outliers.

(1) Keyframe-based detection: By offloading the keyframe information from the agent to the server with a relatively low bandwidth (see Section 4.3), the server is able to receive the visual information from each agent in real time with a low information loss rate. Once the keyframes are received, we perform visual-based loop closure detection, similar to [4], using the bag-of-words place recognition approach DBoW2 [13]. A single database is shared among all agents to enable cross-robot loop closure detection. Loop closure candidates *Q* are first detected by querying the database, and we find the best *N* candidates in *Q* via descriptor-based 2D–2D brute force matching. For matched candidates, we check their associated 3D landmarks, which are reprojected from the candidate frame to the query frame, and vice versa. If sufficient inliers are found via the 3D–2D RANSAC process, we perform an optimization for the corresponding relative pose $T_{ij}$ by minimizing the reprojection error.

(2) Signed Distance Function (SDF)-based check: After the keyframe-based loop closure detection, we obtain an initial transformation $T_{ij}$ between agent *i* and agent *j*. By querying the timestamps of the loop closure, we can find the corresponding TSDF submaps $S_{i}$ and $S_{j}$. Before performing the SDF-based outlier rejection, we need to compute the submap’s isosurface points $P_{S}$ by using the marching cubes algorithm [30] and computing the submap’s Euclidean Signed Distance Function (ESDF) $\phi_{s}$ by propagating the Euclidian distances outside the TSDF, as described in [1].

In each TSDF submap $S_{i}$ and $S_{j}$, the marching cubes algorithm is used to extract points $P_{S_{i}}$ and $P_{S_{j}}$ on the isosurface (i.e., zero level set). The isosurface is defined as follows: (4)ϕ(x,y,z)=0
where ϕ(x,y,z) represents the TSDF value at a given voxel position (x,y,z). The marching cubes algorithm traverses the voxel grid, checking whether the TSDF values of neighboring voxels cross the isosurface ϕ=0, then it generates corresponding triangle fragments at the isosurface to form the point sets $P_{S_{i}}$ and $P_{S_{j}}$.

During loop detection, we align the isosurfaces of the two submaps and evaluate the alignment error. Suppose we have an initial transformation matrix $T_{ij}$ that transforms the isosurface point set $P_{S_{i}}$ to the coordinate frame of $S_{j}$, resulting in a transformed point set: (5)$$P_{S_{i}}^{\prime }=T_{ij}P_{S_{i}}$$

Then, we can calculate the TSDF value $\phi_{S_{j}}\left(p\right)$ for each transformed point $p\in P_{S_{i}}^{\prime }$ in $S_{j}$, forming the SDF error metric: (6)$$d_{\mathrm{SDF}}=\frac{1}{|P_{S_{i}}^{\prime }|}\sum_{p\in P_{S_{i}}^{\prime }}\left|\phi_{S_{j}}\left(p\right)\right|$$

Ideally, if the loop detection is successful, the aligned point set $P_{S_{i}}^{\prime }$ should coincide with the geometry of $P_{S_{j}}$, making the SDF error $d_{\mathrm{SDF}}$ close to zero. To increase robustness, we further introduce a weighted SDF error, taking into account the voxel weights w(p): (7)$$d_{\mathrm{weighted}}=\frac{\sum_{p\in P_{S_{i}}^{\prime }}w\left(p\right)\left|\phi_{S_{j}}\left(p\right)\right|}{\sum_{p\in P_{S_{i}}^{\prime }}w\left(p\right)}$$

If the weighted SDF error exceeds a preset threshold, this loop detection is deemed invalid and is discarded. Otherwise, the loop detection is considered valid.

These two pieces of information are calculated once the server receives the submaps. For a perfect loop closure $T_{ij}$, the isosuface points of $S_{i}$ should always lie on the zero-level set of $S_{j}$, and vice versa. Based on this assumption, we formulate the SDF-based outlier rejection problem as follows: (8)$${\bar{d}}_{iso}=\frac{1}{N}\left(d_{iso,S_{i}}\left(P_{S_{j}},T_{ij}\right)+d_{iso,S_{j}}\left(P_{S_{i}},T_{ij}^{-1}\right)\right)$$
where $d_{iso}$, $S_{i}$ is the sum of weighted SDF values, and *N* is the sum of all weights.
(9)$$d_{iso,S_{i}}\left(P_{S_{j}},T_{ij}\right)=\sum_{p_{iso}\in P_{S_{j}}}w_{s_{i}}\left(T_{ij}p_{iso}\right)\Phi_{S_{i}}\left(T_{ij}p_{iso}\right)$$
(10)$$N=\sum_{p_{iso}\in P_{S_{j}}}w_{s_{i}}\left(T_{ij}p_{iso}\right)+\sum_{p_{iso}\in P_{S_{i}}}w_{s_{j}}\left(T_{ij}^{-1}p_{iso}\right)$$

We calculate the average distance ${\bar{d}}_{iso}$ based on the weighted SDF value $\Phi_{S}$ of the transformed isosurface points $P_{S}$. For a minimum fraction of points both in $P_{Si}$ and $P_{Sj}$, the average distance ${\bar{d}}_{iso}$ should be close to 0. Otherwise, we reject the loop closure $T_{ij}$ and consider it an outlier.

#### 3.2.3. Global Pose Graph Optimization

In order to maintain global consistency across different agents, we perform global pose graph optimization to correct for the drifted poses of the agents. Different from [12], which only performs pose graph optimization based on the relative pose information, we incorporate the dense geometric information of SDF submaps by performing submap matching to further improve the system’s global consistency. Although [12] utilized the SDF submap matching method, it only used it to find loop closures, and the submap matching method is based the time-consuming ICP-based method. Contrary to this [12], we adopt the lightweight correspondence-free submap matching method, as proposed in [9].

On the agent, based on the assumption that the pose estimation errors from the VIO module accumulate slowly over time, we build a series of submaps ${\left\{S_{i}\right\}}_{i=1}^{N}$ at a fixed frequency, and each submap stores the sensor trajectory. Once a submap is sent to the server, we delete it in the agent’s memory and begin to generate the next new submap. We define the pose in the middle trajectory of the submap as the submap pose $T_{WS^{i}}$, and we optimize the submap poses ${\left\{T_{WS^{i}}\right\}}_{i=1}^{N}$ in the PGO module on the server. We solve the nonlinear least squares minimization problem as follows: (11)$$\arg\min_{\chi }\sum {∥e_{rel}^{i,j}\left(T_{WS^{i}},T_{WS^{j}}\right)∥}_{\Sigma_{rel}}^{2}+\sum {∥e_{reg}^{i,j}\left(T_{WS^{i}},T_{WS^{j}}\right)∥}_{\sigma_{reg}}^{2}$$
where
(12)$$\chi =\{T_{WS^{1}},T_{WS^{2}},\cdots ,T_{WS^{N}}\}$$
are the submap poses. ${∥e_{rel}∥}_{\Sigma_{rel}}^{2}$ and ${∥e_{reg}∥}_{\sigma_{reg}}^{2}$ stand for the relative constraints and the correspondence-free registration constraints. Based on the relative pose information, the relative pose constraints can be categorized as odometry constraints and loop closure constraints. The structure of the pose graph across different agents is depicted in Figure 2.

(1) Relative pose constraints: The odometry constraints are constructed for each agent’s consecutive submaps, and the deviation is defined by their odometry-estimated relative poses. Thus, for a consecutive submap pair $S_{i}$ and $S_{i+1}$, the odometry residuals can be formulated as follows: (13)$$e_{\mathrm{rel}-\mathrm{odom}\;}^{i,i+1}\left(T_{WS^{i}},T_{WS^{i+1}}\right)=\log\left({\hat{T}}_{S^{i}S^{i+1}}^{-1}T_{WS^{i}}^{-1}T_{WS^{i+1}}\right)$$
where
(14)$${\hat{T}}_{S^{i}S^{i+1}}=T_{C^{k}C^{k+1}}T_{C^{k+1}C^{k+2}}\ldots T_{C^{k+N-1}C^{k+N}}$$
is the estimated submap relative transformation, which is constructed through the concatenation of VIO poses joined by the sensor frames ${\left\{C_{l}\right\}}_{l=k}^{k+N}$.

The construction of the loop closure constraints follows the same concept as the odometry constraints. After receiving the loop closure inliers, we obtain an estimated transformation ${\hat{T}}_{C^{m}C^{n}}$ from the sensor frame $C^{n}$ to the sensor frame $C^{m}$. By querying the sensor timestamps at $C^{m}$ and $C^{n}$, we can find the corresponding submaps $S_{i}$ and $S_{j}$ that contain the sensor frames $C^{m}$ and $C^{n}$. Thus, we have: (15)$$e_{\mathrm{rel}\_\mathrm{loop}}^{i,j}\left(T_{WS^{i}},T_{WS^{j}}\right)=\log\left({\hat{T}}_{C^{m}C^{n}}{\left(T_{WS^{j}}T_{S^{j}C^{n}}\right)}^{-1}\left(T_{WS^{i}}T_{S^{i}C^{m}}\right)\right)$$
where $T_{S^{i}C^{m}}$ and $T_{S^{j}C^{n}}$ are the poses of the sensor frames $C^{m}$ and $C^{n}$ in their corresponding submap frames.

(2) Registration constraints: Utilizing the dense geometric information of the SDF submaps can further enhance the system’s consistency. To perform correspondence-free submap matching, we first need to detect overlapping pairs of submaps using the Axis-Aligned Bounding Box (AABB) (see [31]). Based on the overlapping pair $S_{i}$ and $S_{j}$, we transform the isosurface points of $S_{i}$ to the submap frame of $S_{j}$. For a perfect alignment, the isosurface points $p_{S_{i}}^{k}$ of $S_{i}$ should always lie on the isosurface of $S_{j}$. The distance between $p_{S_{i}}^{k}$ to the isosurface of $S_{j}$ can be read by the ESDF $\Phi_{S}$ of submap $S_{j}$. Thus, we can formulate the registration constraints as follows: (16)$$e_{reg}^{i,j}\left(T_{WS^{i}},T_{WS^{j}}\right)=\sum_{k=0}^{N_{S_{i}}}r_{S_{i}S_{j}}{\left(p_{S_{i}}^{k},T_{S^{j}S^{i}}\right)}^{2}$$
where $N_{S_{i}}$ is the total number of isosurface points in submap $S_{i}$, and the registration residuals are as follows: (17)$$\begin{array}{cc} r_{S_{i}S_{j}}\left(p_{S_{i}}^{k},T_{S^{j}S^{i}}\right) & =\Phi_{S_{i}}\left(p_{S_{i}}^{k}\right)-\Phi_{S_{j}}\left(T_{S^{j}S^{i}}p_{S_{i}}^{k}\right)\\ \; & =-\Phi_{S_{j}}\left(T_{S^{j}S^{i}}p_{S_{i}}^{k}\right) \end{array}$$
where $\Phi_{S_{i}}\left(p_{S_{i}}^{k}\right)=0$ for all isosurface points $p_{S_{i}}^{k}$ on $S_{i}$, since they lie on the isosurface of themselves, and $T_{S^{j}S^{i}}$ can be represented by the submap poses: (18)$$T_{S^{j}S^{i}}=T_{WS^{j}}^{-1}T_{WS^{i}}$$

The computation time for optimization based on registration constraints is proportional to the number of points on the isosurfaces. The pose graph optimization on the server side will be computationally expensive with increases in the numbers of submaps. To enable real-time performance, we adopt the lightweight optimization strategy based on the subsampling of isosurface points, as proposed in [9]. The subsampling mechanism is proportional to the weight of the isosurface points, which is computed by interpolating the weights of the voxels near the point. As in [9], we set the subsampling rate to 5% to balance accuracy and runtime.

## 4. Results

In this section, our aim is to validate the efficiency, accuracy, and applicability of the proposed dense multi-robot SLAM system under the centralized architecture.

### 4.1. Dense TSDF Mapping

For the evaluation of the dense mapping performance, we utilize two visual-based datasets: the Cow&Lady Dataset [1] and the EuRoC Dataset [2]. Note that since the [2] only provides stereo images, we employ the semi-global matching method [32] to generate the dense pointclouds. The evaluation results are illustrated in Figure 3, and the evaluations are conducted on a PC with an Intel Core i9-12900K CPU. For evaluations on the TSDF mapping performance, we measure the per-frame TSDF update time and the TSDF mapping accuracy. As shown in the second column of Figure 3, our method’s TSDF update time outperforms Voxblox [1] and Voxfield [16], especially for the small voxel sizes. The significant reduction in the TSDF update time is attributed to the early termination of raycasting, as discussed in Section 3.2.1. In terms of the TSDF mapping accuracy, we adopt the non-projective distance proposed in Voxfield to compensate for the projective distance error in Voxblox. As shown in the third column of Figure 3, our method achieves the highest accuracy. While we adopt the same method as Voxfield to increase the mapping accuracy, the grouped raycasting method utilized by Voxfield merges all points in a voxel to a single point, resulting in the loss of more information compared to our subvoxel-based division process, where each voxel contains eight points.

Furthermore, we assess the ESDF error generated by the TSDF map, given that the ESDF map is used in loop closure detection and global pose graph optimization modules on the server side. Note that we adopt the TSDF propagation method to generate the ESDF map, as proposed in Voxblox. This method directly generates ESDF values outside the truncated voxels. As shown in the last column of Figure 3, our ESDF map achieves better accuracy. This is due to the fact that the ESDF map is generated based on the TSDF value. If the TSDF value has a higher error, this error will accumulate through the propagation process, resulting in a higher ESDF error. As a result, this will affect the accuracy of selecting the correct loop closures (see Section 3.2.2) and the accuracy of registration constraints in the PGO module (see Section 3.2.3).

### 4.2. Multi-Robot Dense SLAM

After the evaluation of the single agent’s TSDF mapping performance, we evaluate the multi-robot dense SLAM performance of our system. As each pointcloud is attached to each camera frame, the pose of each frame reflects the quality of dense mapping performance. Thus, we evaluate our system’s estimated trajectory accuracy. For comparison, we compare our system against the state-of-the-art VIO frameworks, VINS-Mono [4] and VINS-Fusion [29], as shown in Table 1. Note that those VIO frameworks do not have the support of the server. In each experiment, we use two sequences of the EuRoC dataset [2] to run on agent 1 and agent 2. The VIO module of our system is VINS-Fusion, and the pointclouds are generated via stereo matching [32]. The voxel size is set to 10 cm. As shown in Table 2, our system exhibits better performance in terms of Absolute Trajectory Error (ATE) compared to VINS-Mono and VINS-Fusion in most cases, and this results in a qualitative reconstruction, as shown in Figure 4. These results demonstrate that our system can efficiently correct the trajectory drift of the agents through the shared information, thus resulting in a better reconstruction of the observed environment.

### 4.3. Real-World Experiments

To evaluate the proposed system’s applicability, we conducted real-world deployment. The real-world system is depicted in Figure 5, comprising two resource-constrained robots and one central server. The hardware setup of our system is shown in Table 2. The VIO module of our system is the GPU version of VINS-Fusion [4,29,33,34], which alleviates the computation pressure on the CPU [35]. The voxel size is set to 5 cm, real environments have many more details, and our voxel size of 5 cm provides much higher accuracy. The communication between the agent and server is through a wireless network. The submap generation and sending frequency is 5 s. In our system design, we adopted a segmented buffering mechanism to alleviate the bandwidth pressure caused by instantaneous transmission peaks. Specifically, we divided the submap data into multiple small chunks of fixed size and transmitted them sequentially, ensuring that the data volume for each transmission remained stable. This approach smooths out data flow, even when a large data volume needs to be sent within each 5-s transmission cycle, thereby avoiding bandwidth peaks. Additionally, the system’s UDP transmission protocol was optimized to ensure that submap data are not lost or delayed during transmission, enhancing the stability and efficiency of data transfer.

We performed the evaluations in the same room of an office building, and the collaborative dense SLAM results are shown in Figure 6 and Figure 7, respectively. The real-time onboard experimental results for the mobile robot are shown in Table 3 and Table 4, respectively. Thanks to the lightweight TSDF mapping method and the submap generation and deletion method on the agent side, the resource-constrained mobile robot was able to perform real-time dense SLAM in a large scenario with relatively low memory storage and a low bandwidth. Our system leverages the decentralized communication capabilities of ROS 2, which uses the Data Distribution Service (DDS) protocol for peer-to-peer communication and distributed node discovery. This enhances the robustness and flexibility of our multi-robot system in dynamic environments.

## 5. Conclusions

In this paper, we propose an efficient centralized collaborative multi-robot dense SLAM system to reduce the computation and memory pressure on resource-constrained mobile robots. To enable real-time dense mapping performance for the agent, we propose a lightweight and accurate TSDF mapping method. On the server, we correlate and optimize the drifted poses of the agents based on both the visual and dense geometric information of the environment. Experiments conducted on pre-recorded datasets demonstrate the efficiency and accuracy of our SLAM system. Finally, the real-world deployment on the mobile robots shows the robustness and applicability of our proposed system.

For future work, we plan to add the path planning module to the agents to enable their navigation capabilities. Furthermore, we plan to develop a centralized multi-robot global planner on the server side to improve the system’s navigation efficiency. Moreover, we intend to expand the system by incorporating more agents to enhance overall efficiency.

## Figures and Tables

**Figure 1 sensors-24-07297-f001:**
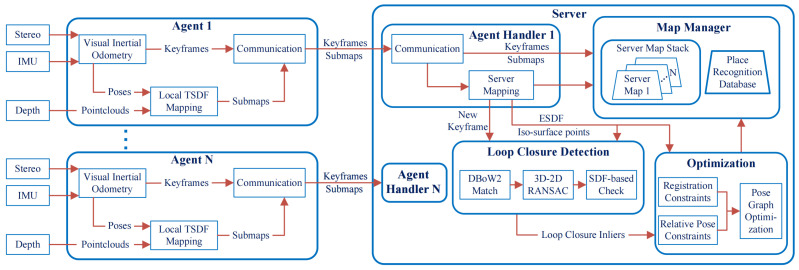
Overview of the SLAM system architecture. Each robotic agent (e.g., a mobile robot) runs real-time visual inertial odometry, maintaining a local TSDF map of limited size and a communication module to send data to the server. The server performs non-time-critical, memory-heavy, and computationally expensive tasks: map management, place recognition, pose graph optimization, and map fusion.

**Figure 2 sensors-24-07297-f002:**
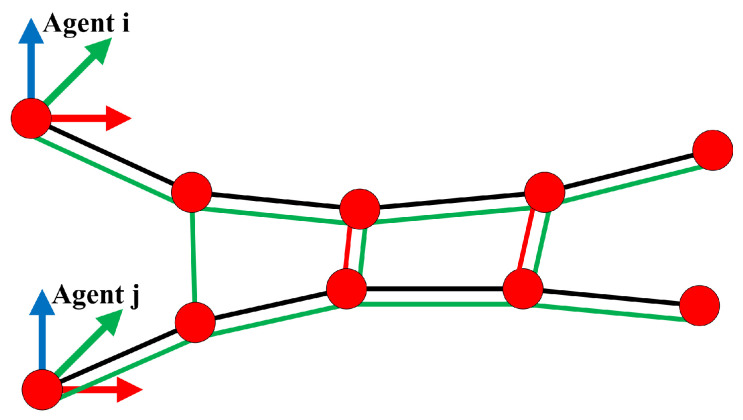
Structure of the global pose graph: the red circles indicate submap nodes (poses), the black lines indicate odometry constraints, the green lines indicate registration constraints, and the red lines indicate loop closure constraints.

**Figure 3 sensors-24-07297-f003:**
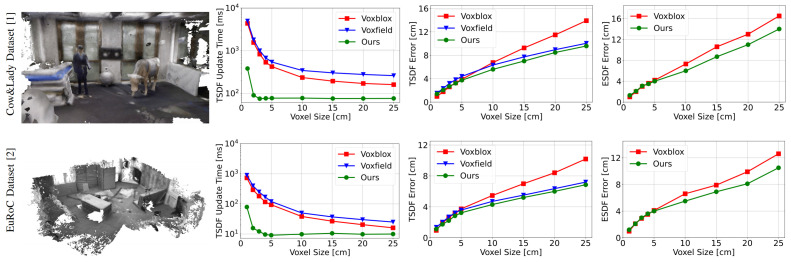
Comparisons of the TSDF mapping performance in terms of TSDF update time, TSDF error, and ESDF error utilizing the Cow&Lady Dataset [1] and the EuRoC Dataset [2]. We compare each method under different voxel sizes.

**Figure 4 sensors-24-07297-f004:**
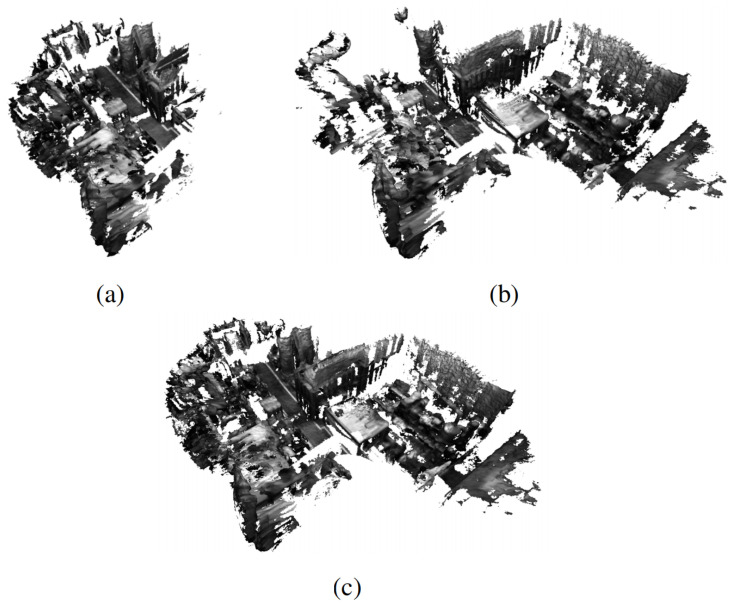
Collaborative dense mapping results of two agents utilizing the EuRoC Dataset [2]. (**a**) Dense mapping result of agent 1 in MH_01 sequence, (**b**) dense mapping result of agent 2 in MH_03 sequence, (**c**) merged global map of MH_01 and MH_03.

**Figure 5 sensors-24-07297-f005:**
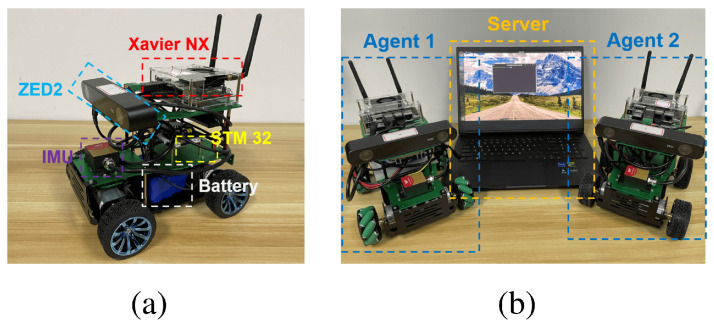
Real-world centralized collaborative multi-robot dense SLAM system. (**a**) The agent, which is a resource-constrained mobile robot. (**b**) The whole system with two agents and one server.

**Figure 6 sensors-24-07297-f006:**
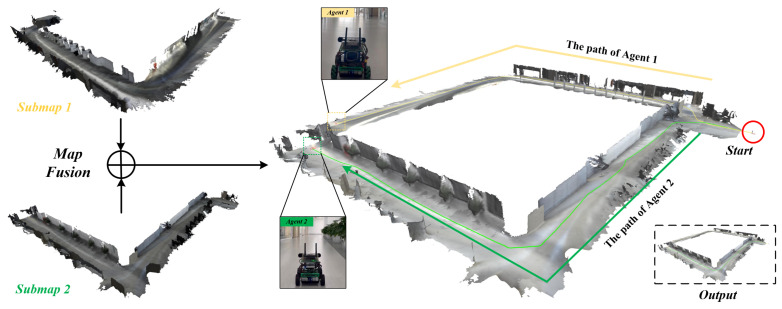
Online collaborative SLAM with two agents (mobile robots) utilizing a centralized architecture in a large office building. The above two pictures depict the SLAM results of a single agent. The right picture illustrates the collaborative SLAM result; the yellow line and the green line represent the trajectories of Agent 1 and Agent 2.

**Figure 7 sensors-24-07297-f007:**
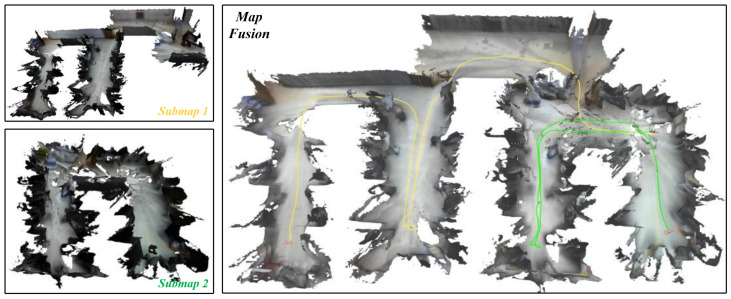
Online collaborative SLAM with two agents (mobile robots) in the same indoor room with obstacles. The yellow line and the green line represent the trajectories of Agent 1 and Agent 2.

**Table 1 sensors-24-07297-t001:** Trajectory ATE comparison.

Seq.	ATE (m)	VINS-Mono	VINS-Fusion	Ours
MH_01 & MH_02	RMSE	0.221	0.247	0.135
	Median	0.181	0.260	0.125
	RMSE	0.178	0.185	0.117
	Median	0.102	0.154	0.106
MH_01 & MH_03	RMSE	0.221	0.247	0.104
	Median	0.181	0.260	0.094
	RMSE	0.228	0.298	0.132
	Median	0.176	0.231	0.095
MH_02 & MH_03	RMSE	0.178	0.185	0.097
	Median	0.102	0.154	0.063
	RMSE	0.227	0.298	0.121
	Median	0.176	0.239	0.081
V1_01 & V1_02	RMSE	0.077	0.117	0.073
	Median	0.058	0.109	0.069
	RMSE	0.090	0.102	0.079
	Median	0.087	0.087	0.073
V2_01 & V2_02	RMSE	0.094	0.117	0.084
	Median	0.070	0.069	0.079
	RMSE	0.118	0.119	0.089
	Median	0.077	0.092	0.072

**Table 2 sensors-24-07297-t002:** Hardware setup for real-world online deployment.

Platform	Type	Characteristics	Sensors
Agent 1	Jetson Xavier NX	1.4 GHz × 6 and 8 GB RAM	ZED 2 Camera (Stereolabs, San Francisco, CA, USA) and WitMotion HWT605 IMU (WitMotion Shenzhen Co., Ltd., Shenzhen, China)
Agent 2	Jetson Xavier NX	1.4 GHz × 6 and 8 GB RAM	ZED 2 Camera (Stereolabs, San Francisco, CA, USA) and WitMotion HWT605 IMU (WitMotion Shenzhen Co., Ltd., Shenzhen, China)
Server	HP Omen 9	5.8 GHz × 24 and 32 GB RAM	-
Router	Mi AX6000	-	-

**Table 3 sensors-24-07297-t003:** Real-world online mobile robot experimental results in the office building.

TSDF Mapping Mean Update Time	TSDF Map Onboard RAM Usage	Mean Keyframe Bandwidth	TSDF Submaps Mean Bandwidth
73.95 ms	3.40%	49.25 KB/s	253.14 KB/s

**Table 4 sensors-24-07297-t004:** Real-world online mobile robot experimental results in the indoor room.

TSDF Mapping Mean Update Time	TSDF Map Onboard RAM Usage	Mean Keyframe Bandwidth	TSDF Submaps Mean Bandwidth
73.86 ms	3.37%	49.23 KB/s	253.09 KB/s

## Data Availability

The data will be shared upon request.

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
