# Peer review of "A Lightweight, Centralized, Collaborative, Truncated Signed Distance Function-Based Dense Simultaneous Localization and Mapping System for Multiple Mobile Vehicles"

_sensors, 2024, doi:10.3390/s24227297_

Round 1
Reviewer 1 Report
Comments and Suggestions for Authors
The concept of using a central server for a multi-robot system with SLAM is not new. For instance, Chebrolu et al. presented a “Collaborative Visual SLAM Framework for a Multi-Robot System” (2015).
The "office building" example is not suitable because the two agents are operating in separate, independent hallways.
To align more closely with the referenced paper, place two agents in the same large room with obstacles. Please illustrate the independent trajectories and maps of each agent in separate figures. Additionally, generate a global map and a combined trajectory from the central server.
Reviewer 2 Report
Comments and Suggestions for Authors
The system emphasizes real-time, resource-efficient operation, addressing the limitations of traditional SLAM approaches in resource-constrained environments, and demonstrating improved performance in both simulated and real-world tests. However, improvement is necessary in terms of clarity, technical explanation, and experimental validation.
1. The author builds dense maps based on the sparse framework CCM-SLAM, and should highlight controbuction on dense mapping. While the author claims a novel combination of keyframe-based detection and SDF-based outlier rejection, the methodology lacks explicit details. In "Signed Distance Function (SDF)-based Check" section, more explanation on how the Marching Cubes Algorithm Algorithm is applied would strengthen the section.
2. The performace is compared regarding TSDF Update Time, TSDF and ESDF Error of different voxel size. Are these metric just a measure of mapping performance, not of tracking accuracy?
3. For tracking accuracy, the keyframes and poses of the agent are only obtained by the odometer, which has nothing to do with TSDF, and the collaboration framework adopted does not involve the feedback from the server side to the agent side, does that mean each agent is just running a separate odometry, and the obtained ATE error is not gained by the synergy frame? If so, the comparison in Table 1 would be meaningless.
4. In the simulation, why the voxel size is set to 10 cm? And in real-world tests, why 5 cm? What is the reason for choosing these Voxel Size?
5. The author mentiond that submaps were transferred every 5s, and evaluated the average bandwidth needed. I wonder if there will be a large instantaneous spike in the transmitted bandwidth in each sub-map transmission cycle? For example, the author mentioned "TSDF Submaps The Mean Bandwidth" is 253 kbit/s, which means that about 1250 kb of data is transferred in a split second during each transfer cycle. If so, is there some buffering mechanism in place?
Comments on the Quality of English LanguageToo many unnecessary Hyphen Usage and extra spaces in the Text.
For example, "Simultane- ous" and "differ ent" in Section Introducion. Please check the full text.
Round 2
Reviewer 1 Report
Comments and Suggestions for Authors
Thank you for the revision. The paper is improved.
In the reply, You mentioned "
|
Decentralized Collaboration Mechanism: Although our framework employs a central server for global map fusion, we designed a decentralized communication mechanism to reduce reliance on a single point. In situations where the server is unavailable or some robots cannot access it, the robots can still collaborate locally, enhancing the system's robustness and flexibility. |
I do not see your "decentralized communication mechanism" being showed in Figure 1 or being discussed in this paper.
Author Response
Comment 1: I do not see your "decentralized communication mechanism" being showed in Figure 1 or being discussed in this paper.
Response 1:
Thank you for your valuable feedback and for pointing out the question regarding the decentralized communication mechanism mentioned in the paper.
To clarify, our reference to a decentralized communication mechanism aligns with the characteristics of ROS 2, the framework used in our system. Unlike ROS 1, which relies on a central roscore for node registration and coordination, ROS 2 employs a truly decentralized architecture facilitated by the Data Distribution Service (DDS) protocol. This enables direct, peer-to-peer communication among nodes without the need for a central node, supporting distributed node discovery and resilient communication.
In practical terms, this decentralized mechanism ensures that even if a central server is unavailable or certain nodes become isolated, the remaining nodes can continue to communicate and collaborate effectively. This feature enhances system robustness and flexibility, allowing our multi-robot system to adapt to dynamic conditions without interruption.
I will revise the paper to better highlight this aspect in 385 to 389 lines.
Thank you again for your feedback, which will help improve the clarity of our work.